# Music Festival Makes Hedgehogs Move: How Individuals Cope Behaviorally in Response to Human-Induced Stressors

**DOI:** 10.3390/ani9070455

**Published:** 2019-07-18

**Authors:** Wanja Rast, Leon M.F. Barthel, Anne Berger

**Affiliations:** 1Department Evolutionary Ecology, Leibniz Institute for Zoo and Wildlife Research (IZW), Alfred-Kowalke-Straße 17, 10315 Berlin, Germany; 2Berlin Brandenburg Institute of Advanced Biodiversity Research (BBIB), 14195 Berlin, Germany

**Keywords:** accelerometry, anthropogenic disturbance, behavioral flexibility, behavior recognition, Erinaceae, non-invasive stress detection, ODBA, urban wildlife

## Abstract

**Simple Summary:**

Mega-events like concerts or festivals can still impact wildlife even when protective measures are taken. We remotely observed eight hedgehogs in a Berlin city park before and during a music festival using measuring devices attached to their bodies. While the actual festival only lasted two days (with about 70,000 visitors each day), setting the area up and removing the stages and stalls took 17 days in total. Construction work continued around the clock, causing an increase in light, noise and human presence throughout the night. In response, the hedgehogs showed clear changes in their behavior in comparison to the 19-day period just before the festival. We found, however, that different individuals responded differently to these changes in their environment. This individuality and behavioral flexibility could be one reason why hedgehogs are able to live in big cities.

**Abstract:**

Understanding the impact of human activities on wildlife behavior and fitness can improve their sustainability. In a pilot study, we wanted to identify behavioral responses to anthropogenic stress in an urban species during a semi-experimental field study. We equipped eight urban hedgehogs (*Erinaceus europaeus*; four per sex) with bio-loggers to record their behavior before and during a mega music festival (2 × 19 days) in Treptower Park, Berlin. We used GPS (Global Positioning System) to monitor spatial behavior, VHF (Very High Frequency)-loggers to quantify daily nest utilization, and accelerometers to distinguish between different behaviors at a high resolution and to calculate daily disturbance (using Degrees of Functional Coupling). The hedgehogs showed clear behavioral differences between the pre-festival and festival phases. We found evidence supporting highly individual strategies, varying between spatial and temporal evasion of the disturbance. Averaging the responses of the individual animals or only examining one behavioral parameter masked these potentially different individual coping strategies. Using a meaningful combination of different minimal-invasive bio-logger types, we were able to show high inter-individual behavioral variance of urban hedgehogs in response to an anthropogenic disturbance, which might be a precondition to persist successfully in urban environments.

## 1. Introduction

Human activity has a significant impact on biological diversity and the persistence of wildlife populations [1]. One example of such human activity is the global process of urbanization, which leads to fast and drastic environmental changes for wildlife [2,3,4]. Some species avoid urbanized landscapes while others thrive and persist in them [5]. Urban areas are not only characterized by an altered landscape but are also hotspots of human activity that may constrain the behavioral repertoire of urban wildlife [6]. A detailed understanding of how human activity impacts urban wildlife populations is essential for conservation and wildlife management and for the resolution of human–wildlife conflicts.

Studying wildlife responses to human disturbance under standardized lab conditions allows inference about causality but often lacks ecological realism. Natural conditions and the complexity of ecological processes are difficult, if not impossible, to integrate into lab studies [7]. In comparison, experimental field studies include a high level of ecological realism, but often cannot identify causality and face many methodical challenges. One challenge is being able to recognize, understand, and clearly distinguish the various environmental factors that affect animal behaviour. Often, there will be a complex set of ecological relationships which cannot be controlled by the researcher. However, recent advances in bio-logging, to remotely monitor animal behaviour and physiology, have removed many of the former limitations of field studies. It is now possible to record the behaviour of free-living wild animals by logging them with high-resolution 3D accelerometers. These data can be used to distinguish different behaviors, as well as to evaluate the rhythmic structure of behaviors.

Behavioral rhythms have evolved as adaptations to the environment and enable organisms to be active at the times most suited to their physiology or ecology. Aberrations from these patterns can result in impairments of fitness. In this study, we focus on a particularly important rhythmic structure, the circadian rhythmicity of activity. Analysis of circadian rhythmicity of activity can be measured using the Degree of Functional Coupling (DFC), which can indicate the disharmony of general behavior patterns caused by stress, illness or disturbances [8,9], and so is well suited to understand the impacts of human activity on urban wildlife.

The European hedgehog (*Erinaceus europaeus*) is protected in many parts of Europe but has experienced serious and continuous declines in the last decades [10,11]. Although hedgehogs originally inhabited rural mosaic structures, they now have higher population densities in urban areas [11,12]. For the protection and management of this species, it is therefore important to assess the adaptive capacity and limits of hedgehogs to urban conditions.

We measured the effects of a music festival in a city park on the behavior of eight urban hedgehogs to investigate the response of hedgehogs to human–wildlife conflict. The biology of the hedgehog can make behavioral studies difficult [13]. They are nocturnal, small and keep hidden during the day, which makes the use of animal-borne loggers all the more valuable. In this study, we work with easy-to-use, non-invasive loggers to infer the stress responses of animals (which are usually estimated by physiological data) by estimating behavior changes in response to a serious stressor through a combination of different loggers and differentiated analysis of the measured data.

We recorded the spatiotemporal behavior of hedgehogs and analyzed it on a fine temporal scale (minutes), before and during a large festival. A music festival in the city is not only a site of human activity but also creates a sudden and drastic change in the environment. Because the festival site had never been previously used for such an event, it is unlikely that the hedgehogs would be accustomed to such a disturbance. We hypothesized that hedgehogs would change their spatial-temporal behavior in response to the festival event. However, behavioral responses are often individual- and sex-specific, which should be considered when studying the effects of anthropogenic disturbances [14,15]. As hedgehogs may adjust their behaviour to avoid contact with human disturbance, we predicted a general decrease in nightly activity area and Degree of Functional Coupling (DFC) during the festival, but individual responses in behavior and activity may vary.

## 2. Materials and Methods

### 2.1. Study Area

Fieldwork was conducted from July to September 2016 within an urban park (Treptower Park) of 88.2 ha, in southeast Berlin, Germany (52.48846° N, 13.46974° E). Treptower Park is open to the general public and contains lawns of short grass (22% of the park area), shrubs of variable density (20.1%), gravel footpaths (36.7%), a playground and a monument site. The park is surrounded by urban pedestrian areas, tarmacked streets and parking areas to the east and south and is bounded by the river Spree to the north and a railway embankment to the west. Within this urban park, streets, fences and the railway embankment impede the movement of hedgehogs. These obstacles can be insurmountable barriers to movement, for example, when fences are installed flush with the ground.

### 2.2. Semi-Experimental Design

We defined the pre-festival phase as 10 August until 28 August 2016, while the festival phase lasted from 29 August until 16 September 2016. The festival phase consisted of three phases: construction, the actual festival and deconstruction. The actual festival, with about 140,000 visitors, took place on 10 and 11 September. The construction of the festival started on 29 August and the deconstruction concluded on 16 September. In total, there were 19 days for the pre-festival phase and 19 days for the festival phase.

During the construction phase, the whole area was fenced, and big mats of aluminium and rubber were placed throughout the park to allow trucks to drive in. Several stands and the two main stages were built in our sample area, which was a 16-ha section of the whole park. The festival ground extended to other parts of the park which were separated from our sampling area by a major four-lane road, which was out of use during the festival. All bushes were fenced to protect wildlife from the festival visitors during the festival phase. During the actual festival, from 10:00 a.m. to midnight, visitors could enter the festival area, and music was played from 10:30 a.m. to 23:00 p.m. on different stages accompanied by light shows. Immediately after the end of the actual festival, the deconstruction of all fences, stages and mats started. A map of the area with marked park, festival and study site boundaries can be found in the Appendix A.

### 2.3. Study Animals and Logger Attachment

At the beginning of August 2016, we carried out two-night surveys at least one hour after sunset to find active hedgehogs by spotlighting (P14.2, LED Lenser, Solingen, Germany). Each hedgehog was marked with five yellow shrink tubes on the spines [16]. The tubes were numbered to allow individual identification during recapture [17].

We equipped 17 hedgehogs with VHF (Very High Frequency) transmitters. On 9 August, we randomly selected eight out of the 17 hedgehogs (four of each sex) and also equipped them with GPS/ACC (Global Positioning System/Acceleration)loggers (E-obs GmbH, München, Germany) using a back plate system with a maximum mass of 30 g, described in detail in [18]. We only used hedgehogs with a body mass greater than 600 g to meet the recommended 5% body mass rule recommended by [19] (Table 1).

Once every week during the study, we recaptured all hedgehogs, weighed them and inspected them for any health problems. On these occasions, we also recharged the data loggers. Nesting behavior was recorded every day by locating the VHF signals of each of the 17 VHF-logged hedgehogs (TRX-1000S, Wildlife Materials Inc., Murphysboro, IL, USA, or Wide Range Receiver AR 8200, AOR Ltd., Tokyo, Japan). On 20 September, we removed all loggers, VHF transmitters and back plates.

### 2.4. Logger Setup

GPS positions were taken from 7:00 p.m. to 7:00 a.m. in 5-min intervals, with bursts of five points to increase accuracy. The VHF transmitters sent signals continuously throughout the whole study period. Acceleration data were recorded alongside the GPS data by the e-obs tags. These accelerometers were programmed to record a short burst of high-resolution data every minute, with a sampling frequency of 100 Hz per axis. All three available axes were measured simultaneously. For individuals 01_2016 and 19_2016, a single burst was 2.64 s long, resulting in 264 measurements per axis. For all other individuals, a single burst was 2.5 s long, with 250 measurements per axis. The burst length is only important for three of the 25 calculated predictors used for the model (see “Section 2.5.2—(1) Behavioral Prediction and Budget” below).

To account for missing data caused by power loss or logger malfunctioning, we removed all dates with less than 1430 measurements taken between 0:00 a.m. and 11:59 p.m. from the analysis of the behavior budget and overall dynamic body acceleration (ODBA) (1440 measurements would be taken for a complete 24 h period). This ensured that only days with the same amount of recordings during both day and night were included, which avoided any bias towards behaviors that only occur during a specific time of day.

### 2.5. Data Analysis

#### 2.5.1. Spatial Data Analysis

Because of high fluctuations in some GPS points, we excluded all points that were more than 1000 m away from the study site. We then calculated the average of all remaining points per time event. Outliers of more than 2 m/s speed from one location to the next were excluded. In a preliminary study, we quantified the mean error of the GPS position to be between 10 m and 40 m, depending on the surroundings. We grouped the remaining points for each night to calculate the used areas from the evening of one day to the next morning.

We calculated the area used by an individual each night and in both phases, pre-festival and festival. We computed the Kernel density estimation 50% (KDE50) for each phase separately using the functions of the adhabitatHR package [20] in R [21] (version 3.5.1, R Core Team, 2018) and R Studio [22]. The KDE50 is used to evaluate the core area used by the hedgehogs per night.

We monitored day nests for all 17 radio-tagged hedgehogs (nine males, eight females, including the individuals from Table 1) from 10 August until 21 September (five days after the festival). We calculated the nest utilization (survival Kaplan–Meier method) period probability using R (package “survival”); we then used the log-rank test (Mantel method, package “coin”) to test the equality of the utilization period distributions between non-festival (before and after) and festival phases for each sex, separately [23].

#### 2.5.2. Acceleration Data Analyses

(1) Behavioral Prediction and Budget

For behavioral prediction, we used a supervised machine learning algorithm which uses data of known behaviors to train and test the model. We took the data of known behaviors used for the model from a preliminary study. In this preliminary study, hedgehogs were logged using the same protocol as the present study and observed in June and July 2016 over several nights, using six hedgehogs (three females, three males) from the same study area in the Treptower Park. In total, four behaviors were considered for the analysis of these animals: resting, defined as not moving regardless of the body posture; balling up, defined as curling up to make a tight ball; walking, defined as slow locomotion; and running, defined as fast locomotion [24].

The data in this preliminary study were recorded in bursts of 2.64 s length with 100 Hz for each of the three axes, resulting in 264 data per burst and axis. To build the model, all six individuals were pooled into one dataset. The model is based on summary statistics calculated from the raw acceleration data using the package accelerateR [25]. For this model, we computed the following set of predictors: mean, standard deviation, inverse coefficient of variation, weighted mean of the autocorrelated power spectrum, variance, kurtosis and skewness, all for each axis separately, and, from a combination of all three axes: q [26], pitch, roll [27] and overall dynamic body acceleration (ODBA) [28].

We chose the Support Vector Machine (SVM) algorithm to classify the predictors for each burst. The SVM represents the predictors in a multi-dimensional space. To separate data points of different classes from each other, a hyperplane is constructed between points of two classes. Points are then classified according to their relative position on the hyperplane [29]. This method was designed to work with binary data. By joining multiple SVMs, it is possible to work with data that have more than two classes [30]. In the present study, we considered three classes: resting, balling up and locomotion. Locomotion includes all behaviors related to relocation (walking and running at various speeds). To account for behaviors that are not included in the model but might occur in hedgehogs, a threshold was set for the SVM. A prediction was only considered reliable when the probability of belonging to a class exceeded 0.7. Otherwise, the behavior was classified as “other” behavior.

The recall (true positives/(true positives + false negatives)) and precision (true positives/(true positives + false positives)) [31] were calculated, as well as the proportion of predictions that were classified as “other”, to evaluate the model after a leave-one-out cross-validation. We used the package “e1071” [32] for the implementation in R.

We prepared the data for the animals of the present study in the same way as data from the model hedgehogs. In addition, we tested the raw data for missing measurements within the bursts. We removed all bursts where less data were recorded than expected under the burst settings. We then used the SVM model to assign a predicted behavior to every burst and its corresponding timestamp. The probability threshold of 0.7 was used to assign the behavior “other” to all bursts that did not exceed the threshold.

(2) Daily Activity Pattern

We calculated the accumulated standard deviation (aSD) by summing up the standard deviation from all three axes for every burst. Using aSD, we calculated the index of diurnality (DI) based on the relative level of activity during daytime compared to night-time for each individual on a given day, with the day starting at civil dawn. We used DI following the methods of [33], in which the different time spans of day and night are taken into account. DI ranges between −1 (absolutely nocturnal) and 1 (absolutely diurnal). We defined civil dawn and civil dusk as the border between day and night; date-specific times for civil twilight were obtained from the National Oceanic & Atmospheric Administration (NOAA, www.esrl.noaa.gov).

Restless phases during the day, such as those triggered by loud music during the festival, would increase the proportion of daytime activity and the DI would thus give an incomplete picture of the influence of the festival on the activity pattern of the hedgehogs. We, therefore, calculated the time span between activity onset and civil dusk (TSdusk). The mean of the aSD was used as a threshold to distinguish between generally active (>mean aSD) and passive (<mean aSD) behaviors.

(3) Overall Dynamic Body Acceleration (ODBA)

The ODBA provides a proxy for energy expenditure [34,35]. At its basis, it is a measurement of general animal body movement, irrespective of behaviour. It was used to map the activity of the animals in order to compare the general activity between the two phases. The ODBA values were taken from the SVM model. To represent whole days, the ODBA values for every 24 h period (0:00 a.m.–11:59 p.m.) were summed up separately for every individual.

(4) Stress Detection

Degree of Functional Coupling (DFC) is a parameter to measure the synchrony of (internal) cyclic behavior and the (external) environmental 24-h period, expressed with a value between 0 (no synchrony) and 1 (maximal synchrony) [8].

We used the aSD to calculate DFCs (see section “Daily Activity Pattern”). Following the protocol of Berger et al., 2003 [8], the time series was auto-correlated in order to filter out noise and enhance rhythmic components and, after a Fourier transform, was used to break the time series down into its rhythmic components, described by the percentage of each component in the original time series. The longest Fourier period tested covers the entire length of the auto-correlation function (here, three days); the shortest Fourier period tested is twice the sampling interval (here, 2 min). The Degree of Functional Coupling is calculated by dividing the Fourier transformation components that harmonize with the 24-h rhythm by the entirety of the Fourier spectrum. To gain an adequate statistical power of the 24-h period, DFCs were calculated for time series of three days using a moving average (first dataset covers days 1 to 3, second dataset covers days 2 to 4, and so on). The resulting DFCs were assigned to the last day of the moving average (day 3 for the first set, day 4 for the second set, and so on).

(5) Statistical Analysis

We compared nightly activity area, behavior, DI, TSdusk, ODBA and DFC between the pre-festival and the festival phases with Linear Mixed Effect Models (LMMs). The nightly activity area, DI, TSdusk and DFC represent daily measurements. The behavior predictions and ODBA were summed up for each day to represent a comparable timeframe. To analyze the change in each parameter, we calculated the differences between each of the 19 daily measurements from the pre-festival phase and each of the 19 daily measurements from the festival phase. We used the same model structure for all parameters (Equation (1)).

diff1 + (1|id)(1)

The diff argument is the measurements of the festival phase subtracted by the measurements of the pre-festival phase. A reduction of the parameter during the festival phase will, therefore, result in a negative difference. We set the animal id as a random factor. The models were fit using the spaMM package [36] in R. We than predicted the mean expected change in the parameter for each individual based on the LMM (predicted data can be found in Appendix A).

## 3. Results

### 3.1. Spatial Results

The size of mean nightly activity area (measured with KDE50) decreased in six out of eight hedgehogs during the festival phase (Figure 1) (for comparison with 95% MCP (Minimum Convex Polygon) and 95% KDE, see Appendix A). The mean expected change in KDE50 and the 95% confidence interval includes the difference of 0 for individuals 02_2016 and 13_2016. The confidence intervals of the remaining six hedgehogs do not include 0. We consider these differences to be significant.

During the pre-festival phase, the probability of a nest being re-used the next day was 66.1% for females and 57.8% for males. During the festival, nests of male hedgehogs were used for a significantly shorter time (Log-Rank, N = 156, Mantel, Z = −2.3327, *p* = 0.02). The probability of using a new nest was ~12% lower in males (57.8% vs. 45.5%). No nests were used for longer than eight days. Values for females were similar in both phases. Differences were, however, not significant for any individual (Log-Rank, N = 88, Mantel, Z = 0.49502, *p* = 0.62) (Figure 2). See Appendix A for raw data of daily nest checks as well as Appendix A for all nest locations.

Daily nest checks showed that individuals 02_2016 and 13_2016 gave birth between 22 August and 28 August, and 29 August and 4 September, respectively.

### 3.2. Model Evaluation for Behavior Prediction

The hedgehog behavior model considered three different behaviors: immobile, balling up, and locomotion. The final dataset consisted of 197 bursts for each of the three behavior classes. The leave-one-out cross-validation showed high values for recall and precision (Table 2). A total of 73 (12%) of the 591 bursts were classified as “other” behaviors using the probability threshold of 0.7.

### 3.3. Behavioral Prediction

Bursts with missing data occurred only in individual 13_2016, where we removed a total of nine bursts. A programming error in the tag on individual 02_2016 led to the removal of five days of data at the beginning of the study (10–14 August 2016). Finally, we removed a total of three days from the pre-festival phase for all hedgehogs except individuals 02_2016 (nine days) and 19_2016 (four days). The number of removed days for the festival phase ranged between 3 and 6 days. The tag of 09_2016 broke down after 12 September 2016, and therefore we removed this individual completely from the analysis as there were only three days with more than 1430 recordings per 24 h of recording in the festival phase. All days removed for all hedgehogs are listed in the Appendix A.

### 3.4. Behavioral Budget

The study hedgehogs differed in their behaviors during the festival phase. Four out of seven hedgehogs reduced the time spent immobile during the festival. Two females (02_2016 and 13_2016) and one male (21_2016) showed no change in time spent immobile, with confidence intervals including 0 (Figure 3A).

Four out of seven hedgehogs changed the time spent in locomotion during the festival. All individuals that changed their locomotion time were female. However, individuals 02_2016 and 13_2016 increased their locomotion time, while 08_2016 and 17_2016 reduced their locomotion time. All analyzed males (01_2016, 19_2016, 21_2016) did not change their locomotion time (Figure 3B).

Five out of seven hedgehogs changed the amount of time spent balled up. One female (02_2016) was the only one to reduce balling up time during the festival. One female (13_2016) and one male (21_2016) did not change their balling up behaviour, while the remaining four hedgehogs increased their time spent balled up. Changes in the “other” behavioral category are shown for completeness but not discussed as the actual behaviors included in “other” are unknown.

### 3.5. Daily Activity Pattern

To fit the LMM, a value of one was added to DI to obtain only positive values, which were transformed using a natural logarithm. All studied hedgehogs were strictly nocturnal, showing negative diurnality indices. Five out of eight hedgehogs reduced their DI (Figure 4). Two males (19_2016 and 21_2016) and one female (17_2016) did not change their DI. 

To fit the LMM, the absolute value of the lowest TSdusk value was added to all TSdusk measurements to produce only positive values. During the festival phase, six out of eight hedgehogs shifted their activity onset to a later time compared to the pre-festival phase, shown by an increased in the TSdusk measure (where zero marks the time of civil dusk, negative TSdusk values represent time before civil dusk, positive TSdusk values represent time after civil dusk). One female (02_2016) and one male (09_2016) did not change their TSdusk values (Figure 5).

### 3.6. ODBA Analysis

To fit the LMM, the ODBA values were transformed into z-scores. Two females (08_2016 and 17_2016) and two males (01_2016 and 21_2016) decreased their ODBA during the festival. One female (02_2016) and one male (19_2016) increased their ODBA, while one female (13_2016) showed no change (Figure 6).

### 3.7. Stress Detection

During the festival phase, six out of eight hedgehogs decreased their DFC. One female (02_2016) increased its DFC and one male (01_2016) did not change its DFC during the festival (Figure 7).

## 4. Discussion

We showed that hedgehogs change their spatial-temporal behavior in at least one of our study parameters during a large disturbance event. Following our predictions, the nightly activity area (KDE50) and DFC decreased for most hedgehogs during the festival. We were able to identify individual changes in the behavioral budget during the festival. We discuss the different parameters below.

### 4.1. Spatial Behavior

We demonstrated a decrease in nightly activity area for most of our study hedgehogs during the festival. Our results are consistent with the recent meta-analysis of [37], reporting a widespread decrease in the mobility of mammals living in highly disturbed environments. They suggested that animals living in built-up landscapes were confined to smaller ranges due to limited movement capacity. During our study, movement limitations set by the park boundaries for hedgehogs were unchanged, thus the decrease in nightly activity area size during the festival phase seems to be an effect of avoidance of the disturbance caused by the festival.

In general, hedgehogs regularly change their nests [13,38]. However, this was the first time a survival analysis was performed on the nesting behavior so there are no relatable data for the pre-festival phase. Building additional nests requires a time investment to find appropriate nesting sites and gather material, which could otherwise be used for foraging. Nevertheless, males changed their nests more often in the festival period. Hedgehogs naturally change their nests regularly, but changing nests more often entails a higher expenditure of energy. Nests used by the females were used longer, which could be explained by the fact that some of them gave birth during the study period and thus were bound to their nests. Changing nests with offspring is even more energetically costly. Highly disturbed mothers might eat their offspring [13] and then change their nest.

### 4.2. Behavioral Analysis

In contrast to spatial behavior, hedgehogs did not change their single behaviors in the festival phase relative to the pre-festival phase in a uniform way. Due to the nocturnal behavior of hedgehogs, any behavioral observation would usually require an observer to be close to the animal to classify behavior. This, however, could already lead to an influence on hedgehog behavior [39]. This study is the first study to remotely record hedgehog behavior, and there are no references as to how hedgehogs behave in the absence of a human observer.

Four hedgehogs reduced the amount of immobile behaviour, while three others showed no change (Figure 3). The interpretation here is difficult because the situations in which a hedgehog may become immobile can be quite different. Hedgehogs will be classified as immobile if they stop walking during foraging or when they are in their nests sleeping during the daytime. The reduction in immobile behavior here could mean a different sleeping posture that is more similar to balling up and therefore treated as such by the SVM. All four individuals that show reduced immobile behavior also showed increased balling up behavior. In addition to these four individuals, one other hedgehog (02_2016) showed an increase in balling up during the festival. Nevertheless, balling up is a defensive behavior which is favored by hedgehogs over moving away from a threat [13]. In a case where construction workers and the music event are perceived as a threat by the hedgehogs, an increase in balling up would be expected and was observed in most of the study hedgehogs.

Interestingly, one female (02_2016) reduced balling up behavior during the festival phase. Direct observation of this individual confirmed that it gave birth during the study period. Judging from the developmental state of the offspring, 02_2016 is estimated to have given birth in the last week before the festival (22–28 August 2016). Having offspring in the nest could prevent the mothers from balling up, either because the nest does not offer enough space or the fact that balling up would prevent the offspring from reaching the teats. The female 13_2016 showed no change in balling up. Visual observations also confirmed that this female gave birth. Judging from the developmental state of the offspring, the offspring was estimated to be born during the first week of the festival (29 August–4 September 2016). The change in balling up behavior was more drastic in 02_2016. Considering the estimated date of birth, the offspring of 02_2016 should have been bigger than the offspring of 13_2016 during the whole festival. This would support the hypothesis of the lack of space in the nest as a reason for reduced balling up. It has to be noted that there are different degrees of balling up. For this study, balling up was defined as a complete ball with no visible head. There are, however, defensive positions a hedgehog can assume where balling is only minimal [13]. This could lead to a misclassification of a defensive behavior as a non-defensive immobile behavior.

A number of mammals have been shown to reduce their movement when in the presence of a predator [40]. In this context, a reduction in locomotion would have been expected during the festival. However, the effect of the festival on locomotive behavior is mixed between the individuals. There were no significant differences among males, suggesting that the festival had no influence on their locomotion. All females, however, showed a significant change in locomotion. Individuals 08_2016 and 17_2016 showed reduced locomotion, while 02_2016 and 13_2016 showed an increase. As the latter two were in lactation they should have experienced a higher energy demand than the females with no offspring [41]. Therefore, they were forced to increase their foraging effort regardless of the festival, while the other two females could avoid taking longer trips.

### 4.3. Daily Activity Pattern

Ordiz et al., 2014 [42], showed that changes in daily activity patterns are useful as a proxy of anthropogenic influences on wildlife. Gaynor et al., 2018 [43], showed that, irrespective of taxa, habitat or location, mammals were more nocturnal in their daily activity patterns in response to human disturbance. For hedgehogs, which are strict nocturnal animals, increased nocturnal behavior is difficult. Indeed, during the festival phase, five out of eight hedgehogs shifted their activity even more into the night shown by a decrease in DI. However, only considering the DI did not effectively show how much the hedgehog’s activity pattern changed during the festival. TSdusk showed that the hedgehogs started their nocturnal activity later during the festival than before the festival. Similarly, Shirley et al., 2001 [44], showed that the Brinkburn Summer Music Festival had a significant effect on the timing of bat behaviour, leaving their priory up to 47 min later on festival nights. Therefore, DI as a solitary proxy for human disturbance is inadequate and should be supplemented by other parameters like TSdusk.

### 4.4. ODBA Analysis

Four out of seven hedgehogs reduced their ODBA during the festival. Two females (08_2016 and 17_2016) reduced their ODBA and also reduced their locomotion during the festival. It could be possible that those two individuals limited their movement in order to avoid contact with the construction works (compare with Section 4.2—“Behavioral Analysis” and [40]). Two males (01_2016 and 21_2016) reduced their ODBA but did not change their locomotion behavior (Figure 3B). Here, it could also indicate that they moved less in their nests during resting periods to avoid being detected in the nest. One female (02_2016) with offspring showed an increase in ODBA. Due to the pregnancy and giving birth, the baseline of the ODBA for this female might have been very low. Only one of the males (19_2016) increased its ODBA during the festival. While it did not change its locomotion behavior (Figure 3B), it decreased its nightly activity area the most (Figure 1). The remaining area might be worse for foraging overall compared to the area usually used, meaning that longer trips to find food might be necessary. Unfortunately, we were unable to differentiate foraging from locomotion behavior within our model. This is most likely due to the position of the acceleration logger on the back of the hedgehog which cannot record movements of the head.

Using ODBA as a proxy for energy expenditure seemed inappropriate in this study. Various circumstances could lead to ODBA values that do not properly reflect energy expenditure. Flexible tendons, different animal gaits and moving up or down slopes have an unknown influence on the ODBA to energy expenditure relationship [28]. In the case of hedgehogs, moving through an area with thick ground cover like poison ivy (*Hedera helix*) or a lot of dead wood is more taxing to hedgehogs than walking over an open field. The additional energy used will not be reflected in higher body acceleration and would lead to an underestimation of the energy expended. Additionally, hedgehogs have the ability to ball up. To hold this position, the animal has to flex muscles which do not result in any body acceleration. This energy expenditure would be completely missed.

### 4.5. Stress Detection

Out of eight hedgehogs, six showed a significant decrease in DFC during the festival, which supports our hypothesis that DFC values were lower during the festival phase. High DFCs are often found in healthy animals or those that are strongly diurnal or nocturnal [45]. Low DFCs indicate that the animal is weakly synchronized with the environmental rhythm, which can be an indicator of stressors or disease, but also parturition [46]. We interpret the changes in DFC as a sign of stress [8]. Individual 02_2016 showed an increase in DFC, as discussed in the Section 4.2—“Behavioral Analysis”; this female was estimated to have given birth in the last week of the pre-festival phase. Parturition was shown to have a great impact on the activity pattern of mothers [46]. Therefore, the DFC in the pre-festival phase is expected to be much lower than normal. Even though individual 13_2016 also gave birth during the study, no pattern similar to 02_2016 was recorded. Individual 01_2016 is the only hedgehog that shows almost no change in mean DFC values. Through our close spatial monitoring, we observed that this hedgehog left the study area and thus the festival grounds.

## 5. Conclusions

Even though urban hedgehogs are expected to show greater tolerance towards anthropogenic activities compared to their rural conspecifics [12], we showed that the music festival had an impact on the behaviour of all study hedgehogs. However, there was no general pattern in the way hedgehogs reacted to the disturbance. Employing different strategies in the same environment was found to have an influence on the fitness of great tits [47].

While looking only at a small portion of the hedgehog population in this park, our study suggests a behavioral plasticity of urban hedgehogs. Behavioral plasticity plays a key role in species adaptation to rapid environmental changes (like urbanization) caused by anthropogenic activities [6], and is also likely to be crucial in the context of coping strategies to human activities. Despite the high plasticity and higher abundance in urban than rural areas, numbers of hedgehogs have decreased across Europe [48,49,50,51,52,53]. Future management of hedgehogs in cities should, therefore, include spatial and temporal protection areas during human disturbances, like festival or park management measures.

We used seven different parameters to measure hedgehog behavior (one using GPS, one using VHF and five using acceleration). We were able to map the behavioral reactions of eight hedgehogs in response to a festival event. The planning of the festival employed wildlife protection measures in building fences and closing the area during the night on the two days music was played. Our results suggest that these measures were insufficient to reduce impacts on hedgehogs. We expect our findings to exemplify responses to disturbance in urban areas. Changes to the environment often happen on a big scale and can occur suddenly. What remains unclear is the potential long-lasting effects of the festival event. Such effects could only be captured by long-term monitoring. The presence of trucks and visitors during the festival might have influenced soil density and the soil fauna, which hedgehogs depend on as a food source. Soil compaction and its effect on hedgehogs have, to our knowledge, not yet been addressed. In a study in the Regent’s Park in London [13], hedgehogs were found to avoid the sports area. Here, it was also suggested that the compaction due to sports activities had an impact on the prey species of the hedgehog, and it has been shown that soil compaction can influence the abundance of earthworms [54], which hedgehogs rely on as a food source. As soil compaction is not quick to reverse, possible long-term effects should be considered in future studies.

Habituation to humans may indeed appear when animals are repeatedly exposed to benign interactions with human activities, although differences exist in the degree to which a species, or an individual, tolerates humans [55,56]. We found a high individuality in our study and therefore strongly recommend that future studies and management plans consider the potential influence of the individuality of solitary species and provide retreat areas [57].

## Figures and Tables

**Figure 1 animals-09-00455-f001:**
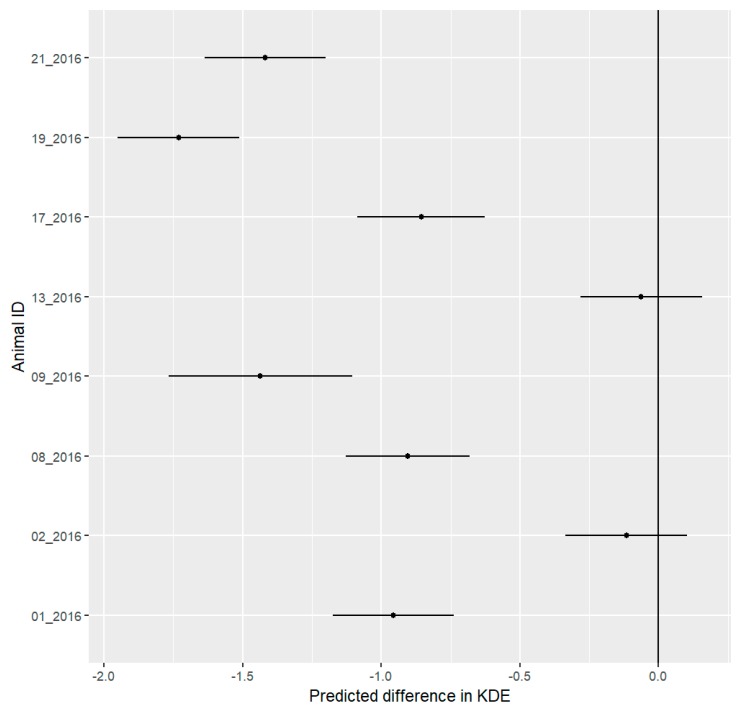
Mean predicted differences with 95% confidence interval for differences in daily Kernel density estimation 50% (50KDE) areas of eight hedgehogs between the pre-festival and festival phases. The predicted difference represents changes in hectares. We consider all differences to be significant where the confidence interval does not include 0. Six out of eight hedgehogs show a significant decrease in KDE50.

**Figure 2 animals-09-00455-f002:**
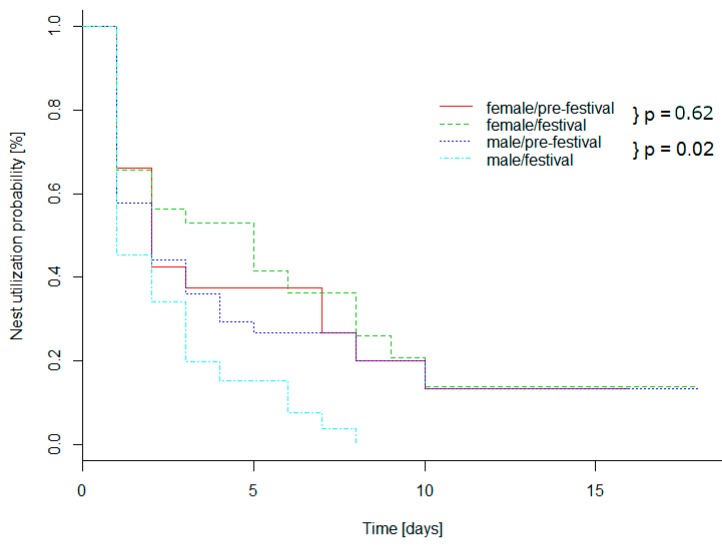
Nest utilization period probability of females and males during pre/after-festival and festival phases, for 17 hedgehogs (nine males, eight females). *P*-values coming from Log-rank test; females: Log-Rank, N = 88, Mantel, *p* = 0.83; males: Log-Rank, N = 156, Mantel, *p* = 0.02.

**Figure 3 animals-09-00455-f003:**
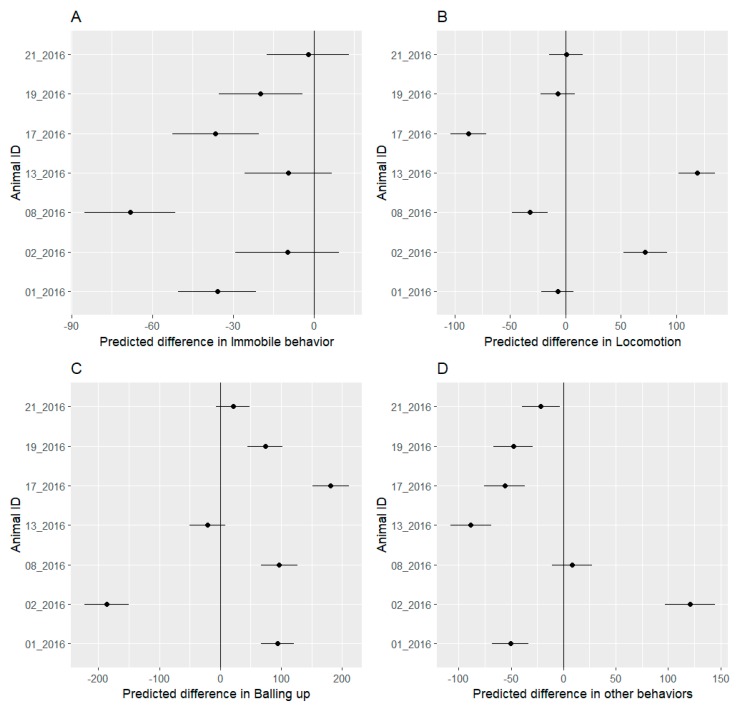
Mean predicted values with 95% confidence interval for differences in daily behavior counts of immobile behavior (**A**), locomotion (**B**), balling up (**C**) and other behavior (**D**) of seven hedgehogs between the pre-festival and festival phases. The predicted difference represents changes in behavior event counts. Negative values represent a decrease in the behavior counts during the festival while positive values represent an increase during the festival. We considered all differences to be significant where the confidence interval does not include 0. Individual hedgehogs showed markedly different behavioral changes during the festival period.

**Figure 4 animals-09-00455-f004:**
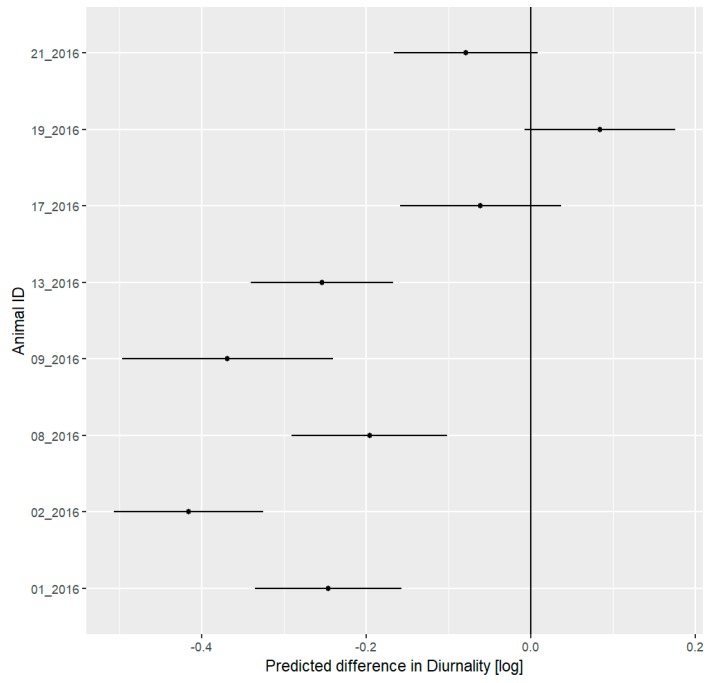
Mean predicted values with 95% confidence interval for differences in daily index of diurnality (DI) of eight hedgehogs between the pre-festival and festival phases. Negative values represent a decrease in the DI during the festival, while positive values represent an increase during the festival. These differences are log transformed. We consider all differences to be significant where the confidence interval does not include 0. Five out of eight individuals showed a significant decrease in the DI.

**Figure 5 animals-09-00455-f005:**
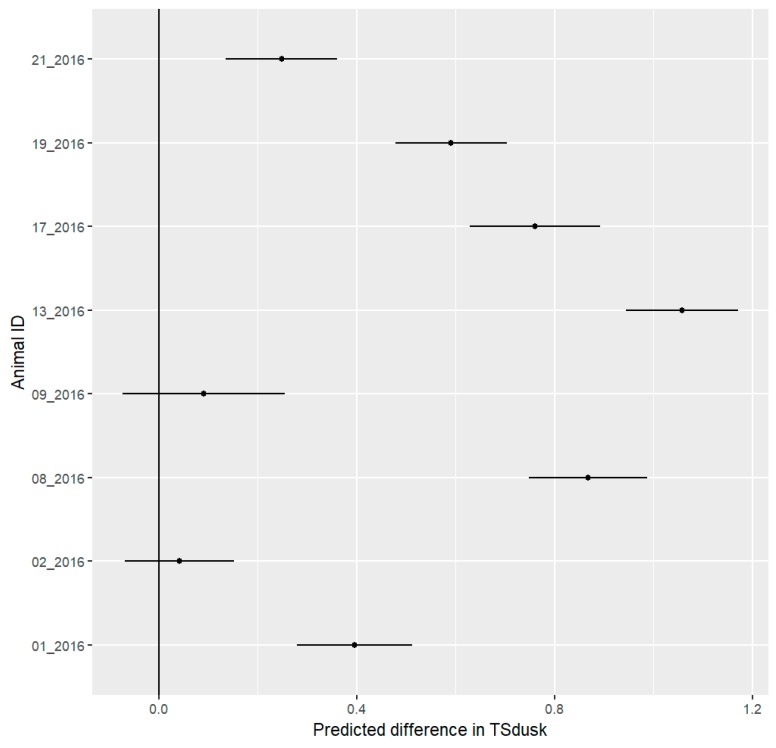
Mean predicted values with 95% confidence interval for differences in daily time span between activity onset and civil dusk (TSdusk) of eight hedgehogs between the pre-festival and festival phases. Negative values represent a shift in activity to a later time of the day during the festival, while positive values represent a shift in activity onset to an earlier time of the day during the festival. We consider all differences to be significant where the confidence interval does not include 0. Six out of eight individuals show a significant positive shift.

**Figure 6 animals-09-00455-f006:**
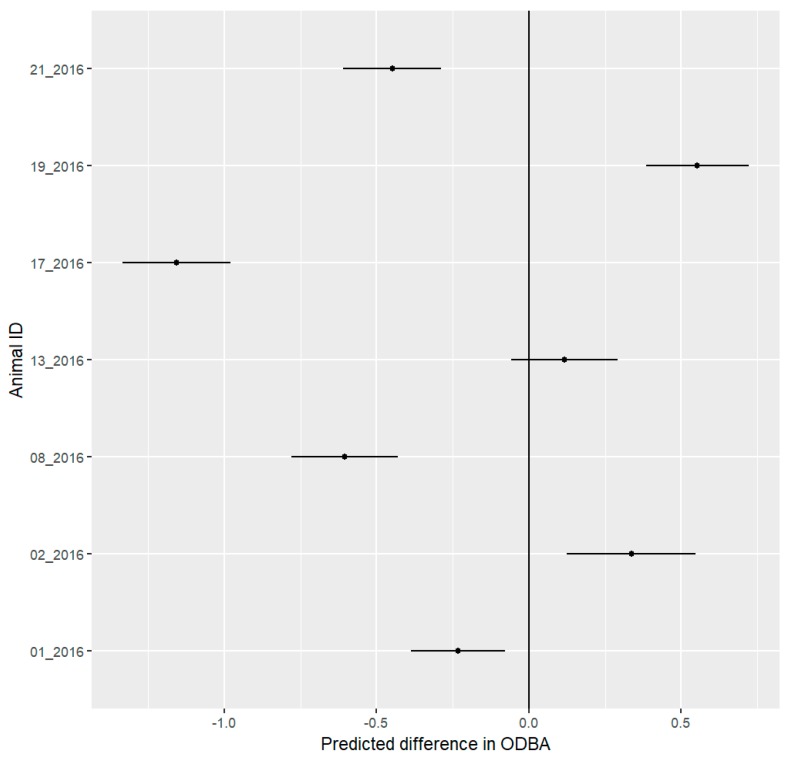
Mean predicted values with 95% confidence interval for differences in daily overall dynamic body acceleration (ODBA) of seven hedgehogs between the pre-festival and festival phases. The predicted difference represents changes in ODBA (values were transformed using the z-score). Negative values represent a decrease in the ODBA during the festival, while positive values represent an increase during the festival. These differences represent changes in the magnitude of standard deviations. We consider all differences to be significant where the confidence interval does not include 0. Four out of seven individuals showed a significant decrease in the ODBA, while two other hedgehogs showed a significant increase.

**Figure 7 animals-09-00455-f007:**
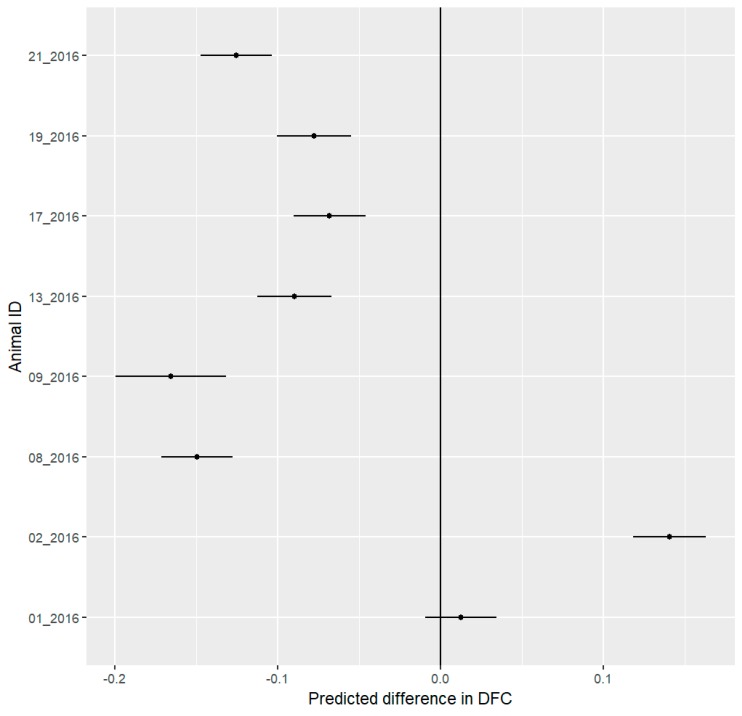
Mean predicted values with 95% confidence interval for differences in daily Degree of Functional Coupling (DFC) of eight hedgehogs between the pre-festival and festival phases. Negative values represent a decrease in the DFC during the festival, while positive values represent an increase during the festival. We consider all differences to be significant where the confidence interval does not include 0. Six out of eight individuals showed a significant decrease in the ODBA, while one other hedgehog showed a significant increase.

**Table 1 animals-09-00455-t001:** Animal identification number (ID), sex and body mass (at the time of logger attachment) of the studied hedgehogs. Abbreviations in the sex column: m = male, f = female.

Animal ID	Sex	Body Mass (g)
01_2016	m	1060
02_2016	f	1090
08_2016	f	795
09_2016	m	830
13_2016	f	725
17_2016	f	1480
19_2016	m	890
21_2016	m	1015

**Table 2 animals-09-00455-t002:** Recall and precision [31] of the hedgehog model.

Behavior	Recall	Precision
Immobile	0.77	0.88
Balling up	0.78	0.90
Locomotion	0.91	0.93

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
