# Peer review of "Music Festival Makes Hedgehogs Move: How Individuals Cope Behaviorally in Response to Human-Induced Stressors"

_animals, 2019, doi:10.3390/ani9070455_

Round 1
Reviewer 1 Report
This is significantly improved on the first submission. Following minor revisions I would recommend this for publication. It will be a good addition to the scientific knowledge base in this area - hopefully others will be able to build on this with a more comprehensive study.
There are still a number of basic mistakes that need rectifying e.g. on lines 96 /97 some of the percentages are to one decimal place, other are not, and some use "." while others use "," as a separator (I do not have the time to point them all out). There are also some grammatical errors that need correcting, I suggest this is proof read by a native English speaker before final submission.
It is good to see this is now clearly written as a pilot paper and the conclusions do not overstate themselves as they did in the first version. I still think that it was a missed opportunity to monitor more hedgehogs; however, I accept there are often limitations e.g. due to funding restraints.
I am pleased to see that the figures have been updated to ain interpretation in greyscale. Supplementary Figure 1: I would like to see differences in GPS fixes between before and during the festival, this could be on the same figure or as two separate figures depending on how easy it is to read.
Given the GPS error rate I agree it would have been problematic to determine which data points were likely to be erroneous. While I am happy with the approach taken it does raise concerns to the robustness of some of the analysis. I am glad to see the inclusion of 95% KDE and MCP, this will help to directly relate the results here to other studies.
It is good to have all of the raw data available in the supplementary, this will allow others to compare results and undertake additional analysis in the future.
Author Response
This is significantly improved on the first submission. Following minor revisions I would recommend this for publication. It will be a good addition to the scientific knowledge base in this area - hopefully others will be able to build on this with a more comprehensive study. \
Response: We agree completely.
There are still a number of basic mistakes that need rectifying e.g. on lines 96 /97 some of the percentages are to one decimal place, other are not, and some use "." while others use "," as a separator (I do not have the time to point them all out). There are also some grammatical errors that need correcting, I suggest this is proof read by a native English speaker before final submission.
Response: We corrected all these minor mistakes and have had another proof reading of a native speaker.
It is good to see this is now clearly written as a pilot paper and the conclusions do not overstate themselves as they did in the first version. I still think that it was a missed opportunity to monitor more hedgehogs; however, I accept there are often limitations e.g. due to funding restraints.
Response: Yes, the financial limitation of the study we see also as a real pity.
I am pleased to see that the figures have been updated to ain interpretation in greyscale. Supplementary Figure 1: I would like to see differences in GPS fixes between before and during the festival, this could be on the same figure or as two separate figures depending on how easy it is to read.
Response: We added two maps in which you can see the GPS fixes during and before the festival period.
Given the GPS error rate I agree it would have been problematic to determine which data points were likely to be erroneous. While I am happy with the approach taken it does raise concerns to the robustness of some of the analysis. I am glad to see the inclusion of 95% KDE and MCP, this will help to directly relate the results here to other studies.
Response: We are also aware of the limitations of our measured data due to the GPS error and the measurement interval and agree with this point.
It is good to have all of the raw data available in the supplementary, this will allow others to compare results and undertake additional analysis in the future.
Response: Thank you very much for your reviewing; it improved our manuscript significantly.
This manuscript is a resubmission of an earlier submission. The following is a list of the peer review reports and author responses from that submission.
Round 1
Reviewer 1 Report
The idea behind this research is good and starts to fill gaps in knowledge which is needed. However, the study design is poor and conclusions often overstate the results. This is a great shame, and with more though this could have been quality research in a needed area. In its current form the study appears to be an undergraduate project reporting a subset of a larger research project or forming a pilot study.
Generally, eight individuals studied for such a small time period in one location and in one year is not sufficient to draw robust conclusions. This work should be re-written and presented as a pilot study, particularly give that the results are inconclusive. On line 411 it is stated that a hedgehog had a nest in a very disturbed area of the festival, this is anecdotal evidence of a lack of impact and little stress. unless there was a reason not mention in the paper one would expect the hedgehog to relocate if it was too stressed, particularly given they have been shown to disperse from unfavourable habitat.
General comments:
· The figures are hard to interpret in black and which, they should be changed to help reader who print in this way.
· Some of the grammar needs checking
· Conclusions not fully supported by the data which is often little more than inconclusive. Often only 5/8 show significance.
· A map of the study site with locations of hedgehogs movement before and during festival is warranted, as with nest sites. Could be in supplementary information. It is highly surprising this is not already included as it will greatly help with the interpretation of the results.
Some more specific points:
· Section 2.1. should include % habitat cover, not just a list
· lines 98/99 state obstacles, this may be true but they are not barriers to movement. This should be made clear to readers. Depending on the construction of the feature (e.g. fence, street) they may pose very little obstacle for hedgehogs, if any.
· Line 123 how were the 8 hedgehog selected out of the 17 - what criteria were used?
· Line 126: Give combined weigh of the kit attached to hedgehogs e.g. VHF, GPS, accelerometer, backpack etc.
· Section 2.4: What was the error rate of the GPS positions
· Line 145: How much data removed, which hedgehogs, which days.
· Section 2.5.1: What about erroneous data points? Using GPS data one needs to check if it makes sense looking at it on a map. Movement speeds differ in different habitats. The GPS point could be within the possible distance moved but is on the other side of an unlikely barrier e.g. river. If there is an obvious movement path, you can get one GPS point that is feasible but unlikely if it implies the animal suddenly detours and then returns to the path.
· line 158: I would like to see 95% MCP and 95% kernel as well, not just 50% kernel
· Line 191: locomotion needs defining
· Line 193: While this is not my area of expertise 0.7 appear to be a fairly low match, is this standard for this work?
· Section 2.5.2: The GPS and accelerometer data should be analysed together to get novel insights and help truth both data sets
· Line 224: ODBA is a poor proxi for daily energy expenditure - which you do say later but be wary about the results and they are presented.
· line 255: what are the P-values? or are these based on no overlap of the CI?
· Line 268: the only ethical concerns here are the handling of hedgehogs with new litters. How did the authors ensure now e.g. abandonment by mother?
· Fig2: Show pooled data. It would also be useful to see the data for each individual, perhaps as a supplementary figure.
· Table 3: incorrectly labelled Table 1
· Line 304: state hedgehog (09_2016) in text, not just from figure 3.
· Figure 4 is hog 13_2016 DFC low because of giving birth?
· Line 344: DFC didn't decrease for all
· Line 354: How many nests we re-used? What was the survival analysis based on, last use of nest or any movement?
· Line 359: This is a leap
· Line 360: Re run analysis excluding the two females that gave birth, do you get a different answer?
· Line 361: Did either female with young move, and if so how long after giving birth?
· Section 4.4. Use the accelerometer data to corroborate your hypotheses here.
· Line 457: What evidence for a miscarriage?
Author Response
Response to Reviewer 1 Comments
The idea behind this research is good and starts to fill gaps in knowledge which is needed. However, the study design is poor and conclusions often overstate the results. This is a great shame, and with more though this could have been quality research in a needed area. In its current form the study appears to be an undergraduate project reporting a subset of a larger research project or forming a pilot study.
This can be indeed considered to be a pilot study as this kind of behaviour analysis was never done before. The nature of festivals in a city makes it very hard to do comparable studies in other populations of hedgehogs or even just replicate the study.
Generally, eight individuals studied for such a small time period in one location and in one year is not sufficient to draw robust conclusions. This work should be re-written and presented as a pilot study, particularly give that the results are inconclusive. On line 411 it is stated that a hedgehog had a nest in a very disturbed area of the festival, this is anecdotal evidence of a lack of impact and little stress. unless there was a reason not mention in the paper one would expect the hedgehog to relocate if it was too stressed, particularly given they have been shown to disperse from unfavourable habitat.
In fact, the sample size is small and general statements about the effects of festivals on the behaviour of hedgehogs - those moreover seem to have different avoiding strategies - has to be criticized. However, under the given conditions we were not able to increase the number of GPS-tagged hedgehogs and there also will be no possibility for a repetition of the festival / study in this park. Nevertheless, the results give a clear picture (see also the statistical relevances) and we think that it is important to make them available to the public. Our experience in examining urban hedgehogs shows that it is becoming increasingly difficult to even find green spaces in the cities where hedgehogs can be found in large (statistically relevant) numbers. In addition, colleagues from London / UK (N. Reeve) have reported that they urgently need information that addresses the impact of music festivals in the city's major parks on the behaviour and, potentially, the population of urban wildlife, as these now at least annually festivals apparently belong to the cultural heritage of big cities becoming bigger and more frequent.
With only 8 individuals, we are able to anecdoticly describe the individually reactions to the festival of each single hedgehog but the aim of our article was to show general reactions and at the same time outlining the great variety of individual reaction possibilities.
General comments:
·The figures are hard to interpret in black and which, they should be changed to help reader who print in this way.
We changed our figures and now they are well interpretable also in black and white.
·Some of the grammar needs checking
A native speaker did read the final version of the manuscript and corrected language mistakes.
·Conclusions not fully supported by the data which is often little more than inconclusive. Often only 5/8 show significance.
We did completely revise the statistical analysis providing other/better significance ratios.
·A map of the study site with locations of hedgehogs movement before and during festival is warranted, as with nest sites. Could be in supplementary information. It is highly surprising this is not already included as it will greatly help with the interpretation of the results.
A Figure with the study site and the nesting sites is placed in the supplementary (Supplementary Figures 1,2)
Some more specific points:
·Section 2.1. should include % habitat cover, not just a list
Proportions of the mentioned habitat types were added to the section
·lines 98/99 state obstacles, this may be true but they are not barriers to movement. This should be made clear to readers. Depending on the construction of the feature (e.g. fence, street) they may pose very little obstacle for hedgehogs, if any.
We tried to make it clearer by adding “embankment can impede the movement of hedgehogs, sometimes these obstacles can be barriers, for example, if fences are installed flush with the ground.”
·Line 123 how were the 8 hedgehog selected out of the 17 - what criteria were used?
There were no further criteria selecting the hedgehogs than the three we mentioned in the chapter: stable occurrence / catching of the individuum on the festival area, weight > 600 gr, sex ratio = 50:50.
·Line 126: Give combined weigh of the kit attached to hedgehogs e.g. VHF, GPS, accelerometer, backpack etc.
We give the most important information of the total weight of the used logger system (VHF, GPS, accelerometer and fixing material) (maximum of 30 g); for further information we refer to the open access publication of Barthel et al. 2018 (https://onlinelibrary.wiley.com/doi/full/10.1002/ece3.4794) which describes the attachment of the loggers (including the single weights) very detailed.
·Section 2.4: What was the error rate of the GPS positions
We did not test the GPS loggers in the study area of the Treptower Park but in a comparable natural setting in Berlin. In this pre-investigation, the mean error of the used GPS-devices and settings (5 measurements per location with a 5 minute interval) was between 10 and 40 m depending on the surrounding.
·Line 145: How much data removed, which hedgehogs, which days.
A complete list was added to the supplementary (Table S3).
·Section 2.5.1: What about erroneous data points? Using GPS data one needs to check if it makes sense looking at it on a map. Movement speeds differ in different habitats. The GPS point could be within the possible distance moved but is on the other side of an unlikely barrier e.g. river. If there is an obvious movement path, you can get one GPS point that is feasible but unlikely if it implies the animal suddenly detours and then returns to the path.
We were sampling every 5 min the GPS location with 5 points that were than combined to one point and afterwards we did the speed cleaning. The mean GPS error of 10 - 40 meters, the relatively small study area, the relatively mobile study animal (max. 2m/s), the measuring interval of 5 minutes and the uncertainness about all possible paths (e.g. holes in the fence) and barriers (e.g. temporary fences, the stages and market stand during the festival) would have made the interpretation of hedgehog movements anyway questionable. Therefore, we wanted to avoid any sort of picking points and used a procedure to make replication and comparison of the both phases as meaningful as possible. It is clear that in our analyses are points that maybe detours but we can not say for sure that there was something that makes the hedgehog detour from its path or the real path is completely different.
·line 158: I would like to see 95% MCP and 95% kernel as well, not just 50% kernel
95% MCP and 95% KDE are now shown alongside 50% KDE in the Supplementary Table 2
·Line 191: locomotion needs defining
Locomotion is now defined as to culmination of walking and running at various speeds.
·Line 193: While this is not my area of expertise 0.7 appear to be a fairly low match, is this standard for this work?
We are unaware of any study using a threshold for the predictions to address the problem of behaviours that are not accounted for in the model. We, therefore, found no references as to how to best determine this threshold. We think it is reasonable to assume that the study hedgehogs do not exactly move like the model hedgehogs (as they are different individuals) so very high probabilities for the predictions would be unlikely. We chose a threshold that is rather low to account for variability of hedgehog movement between individuals but still far higher than a random prediction which would be 0.33 for 3 behaviour categories.
·Section 2.5.2: The GPS and accelerometer data should be analysed together to get novel insights and help truth both data sets
That is a very good point. Associating behaviour to specific locations would enable us to study the behaviour changes in more detail. In our study, however, we think that our study design does not allow a trustworthy associations of behaviour and location. The park has a lot of very small mosaic like shrub patches. With the estimated mean error of between 10m and 40m we would have a high chance of creating false positives of hedgehogs being outside of the shrubs. It could very well be that this type of analysis would be unsuited for a species that inhabits only small areas with a lot of variation.
·Line 224: ODBA is a poor proxi for daily energy expenditure - which you do say later but be wary about the results and they are presented.
We agree on this point. With the reporting of our new results we hope to highlight that increased ODBA values can result from increased locomotive behaviours or restlessness during resting times.
·line 255: what are the P-values? or are these based on no overlap of the CI?
The new reporting on significant results is now based on the CI not including 0
·Line 268: the only ethical concerns here are the handling of hedgehogs with new litters. How did the authors ensure now e.g. abandonment by mother?
We never opened a nest or similar, the observations where by chance because of sightings of the offspring when older or open nest during hot summer days
·Fig2: Show pooled data. It would also be useful to see the data for each individual, perhaps as a supplementary figure.
We added the Raw data to the supplementary because we are not sure what the pooled data mean (Supplementary Figure 2) Here, all individual data can be seen.
·Table 3: incorrectly labelled Table 1
Due to the reconstruction of the analysis there now longer is a Table 3
·Line 304: state hedgehog (09_2016) in text, not just from figure 3.
Due to the reconstruction of the analysis the statement is no longer true and was omitted
·Figure 4 is hog 13_2016 DFC low because of giving birth?
This is possible. However we estimated that the time of birth was at the beginning of the festival. But the time just before giving birth was also shown to lower the DFC in mouflon sheep
·Line 344: DFC didn't decrease for all
We added “for most hedgehogs” to make clear this is not the case for all
·Line 354: How many nests we re-used? What was the survival analysis based on, last use of nest or any movement?
We added the raw data of the nest use in the supplementary (Supplementary Figure 2) were you can see how many nests were re-used by the different individuals. In general, hedgehogs used only one nest per day. We registered these day nests and chronologically numbered them for each individual and study period to use these information for survival analysis.
·Line 359: This is a leap
We changed this statement.
·Line 360: Re run analysis excluding the two females that gave birth, do you get a different answer?
We are not sure what animals rear young during this time as well, thus removing this to animals would not be a proper way to address the problem
·Line 361: Did either female with young move, and if so how long after giving birth?
Considering the nest of the hedgehogs it was mainly observations by chance thus the exact date is just a rough estimate. During the work with hedgehogs we observed that shortly before the females leave the hoglets they move with them to a new location. In one of our cases we had a female move earlier maybe because of the deconstruction of the festival.
·Section 4.4. Use the accelerometer data to corroborate your hypotheses here.
We do not understand this suggestion as section 4.4. is about the ODBA-analysis and its results but ODBA are calculated using the acceleration data (measured by the accelerometers)
-Line 457: What evidence for a miscarriage?
We are not sure if it was a direct loss or in the first days, from outside of the nest she occupied for several days, at one day we heard strange noises and on this and several days afterwards the nest was surrounded by flying wasps, and the female was observed active during the day.
Reviewer 2 Report
This is a fascinating study using GPS and accelerometers to quantify and contrast the behaviour of 8 hedgehogs during a 19 day period before construction activities began for a music festival in Berlin versus a 17 day period when construction work and the festival itself meant a substantial amount of human activity including activity at night when hedgehogs are active. It would have been useful and interesting to monitor ‘recovery’ of natural activity patterns during the weeks after the festival, but the lack of this information is not a major problem.
The results were complex and possibly insightful – in particular, that individuals appeared to differ in their response to the music festival, unfortunately, the statistical analyses were too ‘primitive’ to allow one to draw clear conclusions. That is, while the study itself is interesting and likely to attract attention (indeed, I would not be surprised if it attracted significant media attention or even inspired a Pixar film), but before it can be published in a good journal, the statistical analyses need to be redone. The metrics themselves were reasonably sophisticated thus making it surprising that the authors relied on what I think of as old-style statistics that are not suitable for these data.
Many of the statistical tests involved Wilcoxon rank sum tests. The description of the statistics used for most measures was only given a 2 line description so I am not certain what the authors did, but I am guessing that they took the 19 data points before versus 17 data points during the construction and music festival for each individual and used the Wilcoxon to compare them. However, to my knowledge, the Wilcoxon does not account for repeated measures. Certainly, the 19 estimates for a given metric (e.g., 50% KDE) on 19 days for the same individual are not statistically independent. The norm in the field of studying individual differences in behaviour would use a mixed model to assess whether individuals exhibit consistency in behaviour, whether there is a time period effect, whether other individual characteristics matter, and whether individuals differ in their time period effect. Of course, doing this for only 8 study subjects is not ideal, but having 36 repeated observations per individual is useful. I strongly advise the authors to speak to experts in analyzing data involving repeated observations on the same individuals. Kate Laskowski also in Berlin is a relevant expert.
I am not certain what the authors did, but my guess is that they also neglected to account for repeated measures on the same individuals when they used a Fisher’s Exact Test to analyze the odds ratios for various behaviors for each individual. Because they did not give us any description of this analysis, I was not sure what they did to calculate an odds ratio. It presumably involved the ratio of behavior before versus during the festival, but what did the authors assume to generate confidence intervals for these ratios?
Speaking of confidence intervals, in several figures they show means with confidence intervals but did not tell us what CI they are showing us: 95% confidence intervals?
Finally, while the authors’ main message is that individuals differed in their responses, they barely acknowledged the relevant literature on consistent individual differences in behaviour (aka animal personalities). While their data do not directly relate to the standard personality axes (boldness, aggressiveness, exploratory tendency etc), the larger issue is the general, strong interest in consistent individual differences in behaviour, in particular, in ecologically relevant behaviour. If the authors had incorporated more of this literature in their paper, they would have seen the mixed model statistical methods that are standard in that field. Key, widely cited papers in the animal personality field are by Sih, Bell, Reale and Dingemanse among others. Again, two high-level experts in this field in Berlin are Kate Laskowski and Max Wolf.
Add-on: If possible, it would certainly be interesting to map the GPS-tracked movements on to a habitat map of the park and the areas of highest human activity during construction or during the festival itself. Statistical analyses on how changes in behaviour relate to specific locations would add a major boost to the power of the paper.
Author Response
Response to Reviewer 2 Comments
This is a fascinating study using GPS and accelerometers to quantify and contrast the behaviour of 8 hedgehogs during a 19 day period before construction activities began for a music festival in Berlin versus a 17 day period when construction work and the festival itself meant a substantial amount of human activity including activity at night when hedgehogs are active. It would have been useful and interesting to monitor ‘recovery’ of natural activity patterns during the weeks after the festival, but the lack of this information is not a major problem.
Indeed it would have been interesting to see how quickly or if at all the hedgehogs recover. Here qe did not sample the data.
The results were complex and possibly insightful – in particular, that individuals appeared to differ in their response to the music festival, unfortunately, the statistical analyses were too ‘primitive’ to allow one to draw clear conclusions. That is, while the study itself is interesting and likely to attract attention (indeed, I would not be surprised if it attracted significant media attention or even inspired a Pixar film), but before it can be published in a good journal, the statistical analyses need to be redone. The metrics themselves were reasonably sophisticated thus making it surprising that the authors relied on what I think of as old-style statistics that are not suitable for these data.
Many of the statistical tests involved Wilcoxon rank sum tests. The description of the statistics used for most measures was only given a 2 line description so I am not certain what the authors did, but I am guessing that they took the 19 data points before versus 17 data points during the construction and music festival for each individual and used the Wilcoxon to compare them. However, to my knowledge, the Wilcoxon does not account for repeated measures. Certainly, the 19 estimates for a given metric (e.g., 50% KDE) on 19 days for the same individual are not statistically independent. The norm in the field of studying individual differences in behaviour would use a mixed model to assess whether individuals exhibit consistency in behaviour, whether there is a time period effect, whether other individual characteristics matter, and whether individuals differ in their time period effect. Of course, doing this for only 8 study subjects is not ideal, but having 36 repeated observations per individual is useful. I strongly advise the authors to speak to experts in analyzing data involving repeated observations on the same individuals. Kate Laskowski also in Berlin is a relevant expert.
We followed your advise and spoke to an expert on mixed effect models and revised our statistical analysis completely. We also changed the way we compare the two phases with each other. We now model the differences of our parameters from the two phases. This enabled us to use the same model structure for all parameters. By using the individual as a random effect we also accounted for the repeated measurements. It is true that we ignored the possible temporal autocorrelation. However, we did not find that a model accounting for this performed better for any of the parameters which is why we are still not modelling this aspect.
I am not certain what the authors did, but my guess is that they also neglected to account for repeated measures on the same individuals when they used a Fisher’s Exact Test to analyze the odds ratios for various behaviors for each individual. Because they did not give us any description of this analysis, I was not sure what they did to calculate an odds ratio. It presumably involved the ratio of behavior before versus during the festival, but what did the authors assume to generate confidence intervals for these ratios?
Because we now use the mixed effect model we can use that to analyse the behaviour counts as well. How we analysed the behaviour should now be clearer as we use the same approach for all parameters.
Speaking of confidence intervals, in several figures they show means with confidence intervals but did not tell us what CI they are showing us: 95% confidence intervals?
Yes, the figures before showed 95% confidence intervals. With the revision of our statistical analysis we also created new figures which also include 95% confident intervals. This is now clearly stated in the figure describtions.
Finally, while the authors’ main message is that individuals differed in their responses, they barely acknowledged the relevant literature on consistent individual differences in behaviour (aka animal personalities). While their data do not directly relate to the standard personality axes (boldness, aggressiveness, exploratory tendency etc), the larger issue is the general, strong interest in consistent individual differences in behaviour, in particular, in ecologically relevant behaviour. If the authors had incorporated more of this literature in their paper, they would have seen the mixed model statistical methods that are standard in that field. Key, widely cited papers in the animal personality field are by Sih, Bell, Reale and Dingemanse among others. Again, two high-level experts in this field in Berlin are Kate Laskowski and Max Wolf.
Thanks to you tip we found the paper Dingmanse et. Al (2013) - Quantifying individual variation in behaviour: mixed effect modelling approaches which gave us the needed insight into how these data can be better analysed and ultimately shaped our approach to find a good model.
Add-on: If possible, it would certainly be interesting to map the GPS-tracked movements on to a habitat map of the park and the areas of highest human activity during construction or during the festival itself. Statistical analyses on how changes in behaviour relate to specific locations would add a major boost to the power of the paper.
We added a map of the park with all the GPS positions for our study hedgehogs. We do not have a detailed plan off the construction works but we indicated the festival area that was worked on. The integration of GPS and behaviour data is a very interesting topic and should deficiently get more attention in the future. Unfortunately we think that this type of analysis is unsuited for hedgehogs. In a pre-study we estimated the error of the GPS positions to be between 10m and 40m. The Treptower Park here has a lot of small shrub patches which are much smaller than the estimated error. Attributing specific behaviour to certain location would be quite prone to false positives when these shrub patches are overlooked. This might be a general problem with habitats that have a lot of small mosaic like structures. With more accurate GPS loggers this could become possible.